# Over 90 endangered fish and invertebrates are caught in industrial fisheries

Leslie A. Roberson ⬡ [1,2]✉, Reg A. Watson[3] & Carissa J. Klein[1,2]

Industrial-scale harvest of species at risk of extinction is controversial and usually highly regulated on land and for charismatic marine animals (e.g. whales). In contrast, threatened marine fish species can be legally caught in industrial fisheries. To determine the magnitude and extent of this problem, we analyze global fisheries catch and import data and find reported catch records of 91 globally threatened species. Thirteen of the species are traded internationally and predominantly consumed in European nations. Targeted industrial fishing for 73 of the threatened species accounts for nearly all (99%) of the threatened species catch volume and value. Our results are a conservative estimate of threatened species catch and trade because we only consider species-level data, excluding group records such as 'sharks and rays.' Given the development of new fisheries monitoring technologies and the current push for stronger international mechanisms for biodiversity management, industrial fishing of threatened fish and invertebrates should no longer be neglected in conservation and sustainability commitments.

[1] School of Earth and Environmental Sciences, University of Queensland, Brisbane, QLD, Australia. [2] Centre for Biodiversity and Conservation Science, University of Queensland, Brisbane, QLD, Australia. [3] Institute for Marine and Antarctic Studies, University of Tasmania, Hobart, TAS, Australia. ✉email: leslie.roberson@gmail.com

Seafood is an important source of protein for billions of people globally, with over 80 million tonnes of marine animals taken from the ocean annually for consumption[1]. Fishing, either targeted or incidental, is the primary driver directly causing declines in marine biodiversity[2]. Numerous global and regional-scale initiatives address fishing pressure on marine species, including regional fisheries management bodies, the United Nations Convention on the Law of the Sea and its subsequent agreements, the Convention on International Trade in Endangered Species of Wild Fauna and Flora (CITES), and the Convention on Migratory Species (CMS). Yet, one-third of fished stocks are exploited at biologically unstainable levels[3] and 1 in 16 marine fish species are listed as threatened with extinction by the International Union for the Conservation of Nature's Red List of Threatened Species (Red List)[4].

A great deal of conservation and fisheries management resources have been invested in reducing the impact of fishing on threatened charismatic species, such as dolphins, turtles, and seabirds[5]. While certain populations of threatened fish and invertebrates are closely monitored with fisheries stock assessments, they are treated differently to other wild animals and are, in many cases, permitted to be caught in industrial fisheries regardless of the species' global conservation status. This is unique to marine fish and invertebrates as industrial-scale exploitation of imperilled terrestrial or charismatic marine species is unacceptable from a conservation perspective, even when some populations are considered stable[6,7]. For example, although highly contested, hunting of African elephants (Loxodonta africana)—listed as Vulnerable on the Red List—is allowed for trophies but not for commercial-scale food provision, even where elephants are locally abundant[4,8–10]. Similarly, hunting whales for food is highly controversial, even for species or populations that could likely sustain regulated exploitation[11]. In contrast, the International Game Fishing Association grants licences to target many threatened fish and sharks, including species that are Critically Endangered, which receives relatively little attention[12].

While we have yet to fish a widely abundant marine fish or invertebrate species to extinction, we have fished populations or stocks to local or functional extinctions, such as totoaba in Mexico, sturgeons in Europe, and white abalone in California[13]. Many stock collapses have been small, short-lived species, proving that slow-growing and long-lived animals are not the only ones at risk[14]. Collapses of individual populations do not necessarily precursor species extinction, primarily because there are economic constraints to exploitation of distant or dwindling stocks. However, widespread government subsidies to enhance fishing capacity allow many sectors to operate at economic loss, further threatening declining fish and invertebrate populations[15,16]. Species that span international borders are highly migratory, or exist in areas beyond national jurisdiction where restrictions on fishing are largely voluntary, are at increased risk of extinction even if certain stocks are well managed[17]. Even for distinct stocks of closely monitored commercial species, there is risk of mismatch between management units and biological units that could mask population declines[18,19]. Populations reduced to severely low abundances can take much longer to recover than predicted, and former levels of abundance can become ecologically infeasible[20,21]. Climate change impacts will exacerbate pressures on threatened fish and invertebrates through warming waters, acidification, and loss of critical habitat and prey availability[22].

Several key fishing and seafood importing nations—notably USA and some European countries—have taken important steps to curb overfishing, actively rebuild overfished stocks, and reduce incidental catch of charismatic species[23,24]. However, the global conservation status of commercially targeted fish and invertebrate species is largely overlooked in fisheries management frameworks, which operate at the level of individual stocks or populations[25]. At a global scale, we lack understanding of the magnitude and extent of exploitation of imperilled species, and which fishing and consuming nations are most important for improving monitoring and management of threatened fish and invertebrates. Here, we use Red List assessment information to (1) determine which globally threatened species appear in industrial catch and import records, (2) determine the volume and value of catch and imports of these species, and (3) identify the countries driving catch and imports of imperilled seafood species.

## Results

**Analyses of catch and imports data.** We found 92 globally threatened species (50 teleosts, 39 chondrichthyans, and three invertebrates) in industrial fisheries catch records between 2006 and 2014. One of these species, Atlantic cod (Gadus morhua), has a controversial Red List status and was omitted from the remainder of our analysis[21,26]. The remaining 91 species comprise 1.6% of the total catch volume and 2.5% of the value, estimated from ex-vessel price data (the price fishers receive for their landed catch). The 60 Vulnerable, 20 Endangered, and 11 Critically Endangered species (Fig. 1) have a wide range of body sizes and life history traits, from small and fast growing to large bodied and slow growing. Three wide-ranging teleosts—haddock (Melanogrammus aeglefinus), Atlantic horse mackerel (Trachurus trachurus), and bigeye tuna (Thunnus obesus)—account for 76% of threatened species catch volume and 64% of catch value. Compared to chondrichthyans, teleost species generally fetch higher ex-vessel prices per kg (Fig. 1, Supplementary Table 1). However, mean price is less meaningful for chondrichthyans because they are often disaggregated with the liver, skin, gills, and especially the fins sold separately at a higher price per kg than the meat[27].

We explored the threats data from the Red List assessments and found that fishing is listed as an ongoing threat for 87 (96%) of the threatened species, and is the only ongoing threat listed for the majority of species (Supplementary Tables 1 and 2). Large-scale, targeted fishing is specifically listed as a threat for 65 (71%) species and is the only ongoing threat listed for seven species: rock grenadier (Coryphaenoides rupestris), sky emperor (Lethrinus mahsena), golden threadfin bream (Nemipterus virgatus), common spiny lobster (Palinurus elephas), and the Southern, Pacific, and Atlantic bluefin tunas (Thunnus maccoyii, T. orientalis, T. thynnus). The global population trend is decreasing for 80 (88%) of these species and the remainder have unknown population trends.

Industrial catch of threatened species can be targeted or incidental (bycatch)[28,29]. To indicate which threatened species are targeted in industrial fisheries, we used the RAM Stock Legacy Database, which compiles stock assessment results for commercially exploited marine fish and invertebrates around the world (https://www.ramlegacy.org/). We found 34 (37%) of the threatened species listed in the RAM database (Fig. 1). These commercially targeted species account for 88% of the threatened species catch volume. Industrial targeting of additional species not listed in the RAM database is indicated by records of international imports in the trade database (four species), and by the IUCN threats data (35 additional species with targeted large-scale fishing listed as a threat). Together, the 73 species account for 99% of threatened species catch volume.

To estimate the final destination of the seafood, we used a global seafood database that uses FAO FishStat Exports and UN ComTrade data to build a virtual marketplace that links fisheries catch to importers and re-exporters[30]. We found species-level

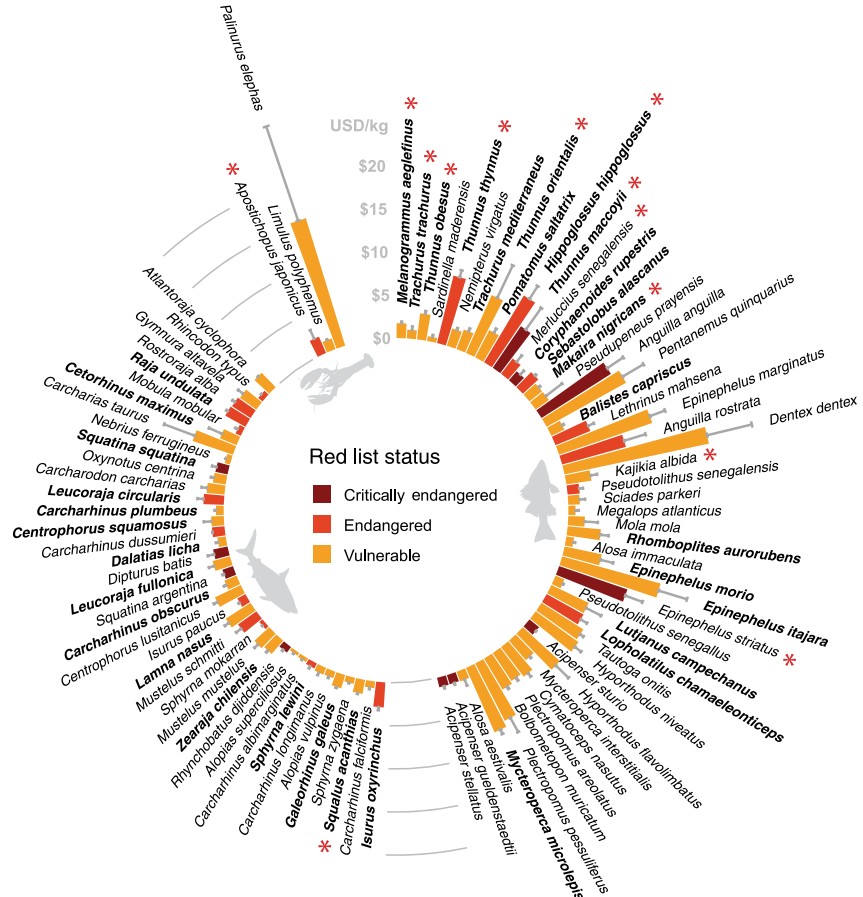

**Fig. 1 Average ex-vessel price and Red List status for 91 threatened catch species from 2006 to 2014.** Prices are global averages for 2010. Error bars show max price for 2010. Species are ordered clockwise by descending catch volume for each taxonomic group (teleosts, chondrichthyans, and invertebrates). The 13 species with red asterisks are found in global import records from 2006 to 2015. The 34 species in bold have commercially exploited populations listed in the RAM Legacy Stock Assessment database. The animalfish silhouettes are from Freepik.com.

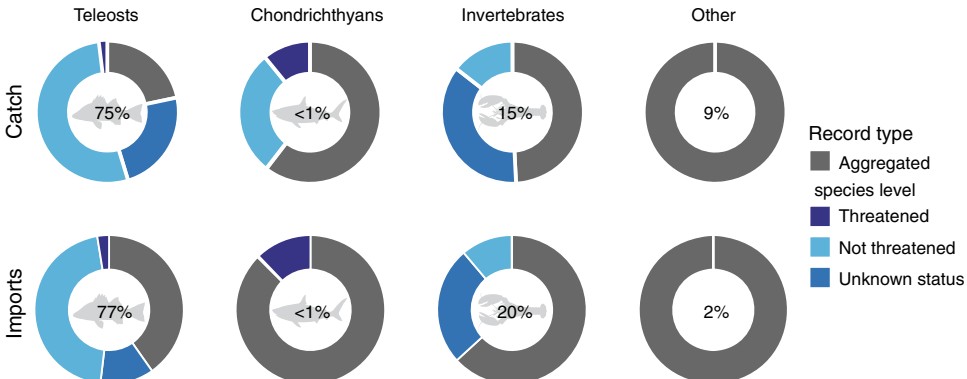

**Fig. 2 Taxonomic resolution of catch and import records.** Proportions of catch and imports volumes recorded at species level are shown in blue and aggregated records are shown in grey for teleosts, chondrichthyans, invertebrates, and other commodities (e.g. "marine animals"). The number indicates the proportion of total catch or import volume in each taxonomic group over the time period (2006–2014 for catch and 2006–2015 for imports). Threatened: Critically Endangered, Endangered, or Vulnerable, Not Threatened: Least Concern or Near Threatened, Unknown status: Data Deficient or has not been assessed, Aggregated: not a species-level record. The fish silhouettes are from Freepik.com.

import records for 13 of the 91 species (11 teleosts, 1 chondrichthyan, and 1 invertebrate, Fig. 1), comprising 2.1% of global import volume and 2.5% of import value (based on ex-vessel prices) from 2006 to 2015. The top three species bycatch volume (Atlantic horse mackerel, haddock, and bigeye tuna) comprise 92% of the total threatened species import volume.

**Resolution of seafood data**. We make a conservative estimate of the volume and value of threatened species catch and imports by limiting our analysis to species-level records. We gauge the extent of our underestimate by comparing species-level to aggregated records (Fig. 2). One-third (33%) of the reported industrial catch volume from 2006 to 2014 consists of aggregated records such as

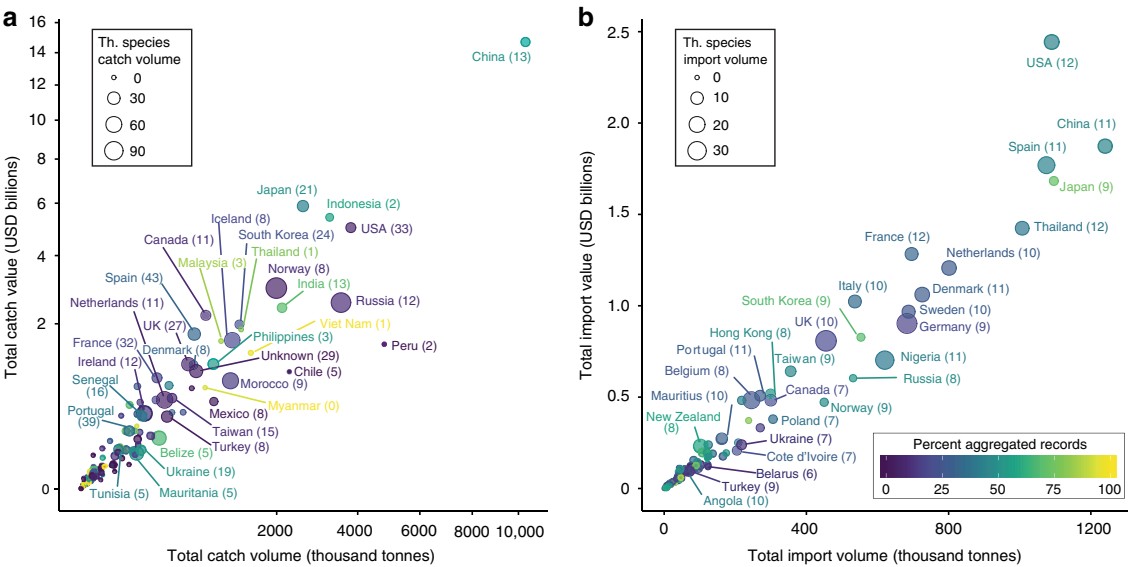

**Fig. 3 Threatened species catch and import volumes and values compared to country totals.** Catch volume and estimated value for 163 fishing countries are shown on a log transformed scale (**a**) and import volume and estimated value for 204 importing countries are shown on a continuous scale (**b**). Bubble size corresponds to volume of threatened species catch or imports (thousand tonnes). Number of threatened species each country catches or imports is in parentheses. Colour shows the percent of each country's catch or import volume that is aggregated (i.e. yellow indicates catch and import volumes mostly recorded in aggregated groups and purple indicates catch and import volumes mostly recorded to the species level). Volumes and values are weighted moving averages for 2014 for catch and 2015 for imports.

"Marine pelagic fishes". Almost one-quarter (23%) of the catch volume is comprised of species that are Data Deficient or have not been evaluated on the Red List. Resolution of catch and import records is much better for teleosts and invertebrates than for chondrichthyans, which have more complete Red List coverage but the largest proportion of aggregated records (Fig. 2). As expected, import records were lower resolution than catch records, with almost half (46%) the total import volume recorded in aggregated commodity groups.

**Country level patterns in catch and imports**. We found records of the 91 threatened species in catch data from 138 of the 163 fishing countries between 2006 and 2014. On average, these countries catch seven threatened species with Spain, Portugal, and USA catching the highest number (43, 39, and 33 species, respectively). The world's major fishers in terms of catch volume and value were not necessarily the countries catching the largest volumes of threatened species (Fig. 3a). Six of the ten countries with the highest volume and value of threatened species catch are European (e.g. Norway, Russia) (Fig. 3a, Supplementary Table 3). However, several countries known to catch threatened species, especially chondrichthyans, have no records of threatened species in the catch database (e.g. Oman, Hong Kong)[31]. Also absent were countries severely lacking fisheries management capacity (e.g. Eritrea, Yemen)[31] or transparency (e.g. Myanmar, North Korea)[32].

Over the decade, 204 countries reported imports of 13 globally threatened species (Fig. 3b). On average, countries importing threatened species imported six of the 13 species. European countries (e.g. Germany, UK, Spain) and USA comprise most of the top importers of threatened species by volume and value, with Nigeria, Thailand, and China also ranking among the top ten (Fig. 3b, Supplementary Table 4). Countries with few species-specific records compared to aggregated records likely catch or import more threatened species than appear in the data (e.g. Myanmar, Malaysia, Philippines, Japan, and South Korea, Fig. 3).

We used linear models to test whether large volumes of threatened species catch or imports were artefacts of good record

keeping (more species-level records) or were simply the countries with the largest volumes of catch and imports. Large volumes of threatened species catch were negatively correlated with larger volumes of aggregated records and positively correlated with larger total catch volumes and with higher per capita GDP, which could indicate greater capacity for catch documentation (df = 139, adj. $R^2 = 0.21$, $p = 0.0015$, $p = 7.4e{-}6$, and $p = 0.0017$, respectively) (Supplementary Table 5). Volume of threatened species imports showed strong positive correlation with total import volume and strong negative correlation with volume of aggregated import records (df = 206, adj. $R^2 = 0.66$, $p < 2e{-}16$), but not with GDP (Supplementary Table 6). The model explained more of the variation in volume of threatened species imports compared to the model of catch volumes, which is not surprising given the much greater variability in catch volumes and record quality between fishing countries compared to importing countries (Fig. 3). Many fishing countries deviate from the pattern of more catch and better records corresponding to larger volumes of threatened species; for example, Peru and Chile, which catch large volumes of least concern anchovy and sardine species in relatively selective fishing gears (Fig. 3a). In contrast, there are fewer records of threatened species imports and poorer record quality overall, thus seafood importers tend to have threatened species imports that are more proportional to their total import volumes (Fig. 3b). Composite governance score was not a significant predictor variable for catch or imports, likely because fishing threatened species is not illegal and there is no binding international requirement to report catch or imports of fish or invertebrate species in high taxonomic detail.

**Discussion**

The 2019 Global Assessment by the Intergovernmental Platform on Biodiversity and Ecosystem Services emphasizes that exploitation is the primary direct driver of marine biodiversity declines[2]. The prevalence of fishing—and targeted industrial fishing specifically—in the Red List data further indicates the importance of controlling large-scale exploitation to ensure the future viability of these species. For the first time, we analyse

industrial fishing data to determine how much and which type of threatened species are reported in catch records and by whom; information critical for focusing conservation and management action towards threatened marine fish and invertebrates.

We present the most conservative estimate of catch volumes of threatened seafood species by excluding unreported catch, records from non-industrial sectors (which are often not reported to the FAO), or catch reported in aggregated commodity groups. Stock assessment and Red List data suggest that most of these threatened species are targeted to some extent in industrial fisheries. Other threatened fish and invertebrate species were undoubtedly caught in industrial fisheries but were not recorded to the species level. For example, many species of sea cucumbers are fished commercially and listed as threatened on the Red List[33], but the Endangered Japanese spiky sea cucumber (*Apostichopus japonicus*) was the only species that appeared in our global catch data. In addition, there were 444 species in the catch records that were Data Deficient or unassessed on the Red List. Models of extinction risk suggest that up to one-quarter of these unassessed marine species may be threatened[34,35]. The number of Data Deficient or unassessed invertebrate species is particularly concerning because invertebrate fisheries are rapidly expanding as market demand grows and many fish stocks decline[36].

Global catch and import records for industrial fishing indicate that European countries play a central role in driving exploitation of threatened fish and invertebrates. However, developed countries with greater monitoring and management capacity (e.g. UK, Norway, Netherlands) tend to have higher resolution catch and import records, which likely results in more records of threatened species compared to countries with few species-level records (e.g. Myanmar, Thailand, Malaysia). We also identify countries that have poor catch and import documentation despite having the financial means for better monitoring (e.g. China, Spain, Japan).

Compared to catch, it is more difficult to identify the countries driving threatened species imports because of the overall lower taxonomic resolution of global seafood trade records. For example, USA has very little industrial reported catch that is not recorded at species level, but almost half of its imported commodities are aggregated records because, like many wealthy nations, it imports seafood from countries with less stringent regulations or management capacity[37]. We likely underestimate the value of imports for wealthy countries and overestimate those of poorer countries because we use ex-vessel prices to compare the value of seafood imports. In general, wealthier countries import more expensive commodities, so the actual value of their imports will be higher compared to lower-income countries importing the same species or commodity group[30].

Ideally, consumers should be able to purchase seafood that is from a well-managed stock that is secure on a global scale, consistent with World Trade Organization measures relating to the conservation of exhaustible natural resources, international fisheries agreements such as the UN Fish Stocks Agreement, and global targets for biodiversity such as the UN Sustainable Development Goal 15[23,38]. Some distinct populations of globally threatened species may be fished sustainably, but the current structure of the seafood supply chain makes it difficult for consumers to make informed, sustainable purchases[38,39]. A crucial first step to better management of fishing pressure on threatened marine species is better taxonomic resolution of catch and trade data, so that we can more accurately understand what species we are catching and consuming and their conservation statuses. Better catch records will also facilitate more accurate Red List assessments[40,41] and help identify marine species that merit consideration of CITES or CMS listings, which aim to better monitor and manage international trade. Although a large proportion of teleost species are listed as Least Concern of extinction,

many species have only been recorded a handful of times, especially those inhabiting international waters where fisheries are least restricted[17].

Governments and fisheries management organizations have made considerable progress in managing fishing and trade of charismatic marine species such as whales and sea turtles[5], but we maintain a cognitive dissonance with threatened fish and invertebrates that we eat. Some fishing sectors have national catch restrictions for certain endangered species, usually for large chondrichthyans caught primarily as bycatch (e.g. basking shark *Cetorhinus maximus*)[4,38]. However, the US Endangered Species Act is the only national legislation that effectively extends beyond direct exploitation of species within domestic borders to address imported species[42]. Threatened seafood species also receive limited international protection from agreements such as the CMS or CITES, which address but do not always restrict international trade, do not restrict catch, and only apply to voluntary signatory countries. None of the 13 internationally imported threatened species from our data are listed on these two conventions (Supplementary Table 1), although many meet the criteria as endangered or migratory species. Atlantic bluefin tuna (Endangered) was denied CITES listing in 2010 after fierce resistance from Japan and other wealthy countries with tuna fleets; the Vulnerable piked dogfish (*Squalus acanthias*) was also denied listing, and the Critically Endangered Southern bluefin tuna has never been nominated[43,44]. Ultimately, voluntary international agreements such as CITES will offer limited protection to imperilled species, unless the signatories shift their focus from purely economic interests to the long-term viability of marine species. Expanding the scope and power of international agreements, such as the recent negotiation of a legally binding instrument for biodiversity beyond national jurisdiction, could potentially be a major gain for threatened fish and invertebrates[17].

Despite the challenges of improving traceability of species across the seafood supply chain, it is increasingly possible and cost effective to identify an animal and trace it to the consumer using emerging technologies such as electronic monitoring, DNA testing, code tags, blockchain, data mining, and artificial intelligence[1,45–47]. For example, OpenSc—one of several new digital platforms for tracing food—has been successful in pilot projects for tuna and Patagonian toothfish[48,49]. Greater and more coordinated efforts from governments, seafood companies, and NGOs are necessary to implement catch documentation schemes, align processes across supply chains, and develop better incentives to improve traceability[46,50].

A few glaring regulatory loopholes remain that impede traceability of threatened species, and seafood in general. One major problem is lack of mandatory reporting of species not listed as targets, as many species are caught intentionally and incidentally in different contexts[28]. Fisheries management often lags behind evolving patterns of targeting as changing resource availability shifts species from bycatch to targets[29]. A second example is the common practice of transshipment—where catch is transferred from a fishing vessel to a cargo vessel (reefer) at sea—often beyond national jurisdiction and enforcement systems[51]. A third key problem is flags of convenience—vessels registered under flags of countries not affiliated with the owner—which typically have lax regulation or enforcement[51]. For example, Russia and Belize both have very high reported catch volumes of the 91 threatened species in our databases, but are well-known flags of convenience for both fishing and reefer vessels, so much of that catch is probably taken and traded by foreign-owned ships[51].

Major fishers and seafood consumers such as China, Japan, USA, and European nations have power and responsibility to improve traceability and sustainability of seafood globally[52], and are also important for reducing industrial fishing impacts on

threatened species. Our analysis also highlights several countries that are not among the world's top fishers or seafood consumers but are particularly important for threatened species. These countries either have large recorded catch or imports of threatened species (e.g. Morocco, Germany) or very low-resolution records (e.g. Myanmar, Malaysia), which may mask high incidence of threatened species. Importantly, the global catch and imports data is recorded at the country level, but a relatively small number of transnational corporations actually do the fishing, processing, and trading[53]. The countries that license these companies to fish in their waters or consume their seafood products can pressure seafood companies to improve production practices. Regional fisheries management and non-governmental organizations both play important roles in persuading and incentivizing countries—and the seafood companies they authorize—to perform better.

Here, we provide the most conservative inventory of global catch and imports of threatened fish and invertebrates as a basis to prioritise research and policy development at the international level. Greater awareness of the global conservation status of seafood species from seafood consumers, fisheries management institutions, and conservation organizations would help expand these initiatives to commercially exploited species of conservation concern. Efforts to preserve marine biodiversity and maintain viable ecosystems will fail if we focus only on charismatic species or individual stocks. We need to treat fish and invertebrates as wild marine animals as well as seafood commodities, better align conservation assessments and fisheries management frameworks, and reduce fishing pressure that is pushing species towards extinction.

## Methods

**IUCN Red List**. We explored the IUCN Red List conservation statuses of all seafood commodities in two global catch and trade databases. We used the Red List because it is the most commonly used global dataset for identifying the types of threat and levels of extinction risk to marine species, it incorporates fishery stock assessment information where available, and typically aligns with fishery management statuses where populations listed as threatened are usually below target fisheries reference points for stock biomass or target catch[19,31,41,54–56]. However, we acknowledge two issues with Red List assessments of some commercially targeted species. First, the global status does not capture the heterogeneity of distinct populations, which is substantial for some species (e.g. Atlantic cod). Second, the Red List's population reduction thresholds were originally designed for terrestrial species, and may overestimate the extinction risk of abundant and fecund species such as tuna and sardines[21,26,57].

We selected all marine invertebrates, teleosts, and chondrichthyan species from the Red List version 2019.2 and matched to the commodity list using species names. We included synonyms and defunct names provided by IUCN. We considered only the global Red List assessments—excluding regional assessments—for three main reasons: (1) regional assessments are disproportionately available for Europe and North America, (2) there is often uncertainty about the congruence between biological populations and management units, and (3) for many species it is not possible to accurately determine which population the catch originates from the global catch data[18]. We made an exception for Atlantic cod, where we used the 2013 European assessment (Least Concern, population trend is increasing) because the 1996 assessment of Atlantic cod as globally Vulnerable was highly controversial[21,26]. Stocks in North America remain depleted after a dramatic crash in the 1980s and the vast majority of the global catch of Atlantic cod now comes from Europe, although there remains some concern about population declines and potential overexploitation of the European cod stocks[58].

We explored the Red List information on threats to the 91 threatened species recorded in the catch and imports data, excluding threats not listed as "Ongoing". We divided the threats into six categories based on the IUCN threats classification scheme, recognizing that the scale of the fishing (e.g. industrial versus small scale) is difficult to define: (1) targeted industrial fishing, (2) incidental industrial fishing, (3) targeted non-industrial fishing, (4) incidental non-industrial fishing, and (5) unspecified fishing. Any threat other than fishing (e.g. pollution, climate change, intrinsic characteristics) we categorized as (6) other (Supplementary Table 2).

**Global catch and imports data**. We linked the Red List information to species-level records in global catch and trade databases to estimate the volume and value of reported threatened species catch and imports from industrial fishing, relative to total catch and imports.

We used the Sea Around Us (SAU) global catch database[59] to calculate the total and average annual catch volumes for each wild-caught marine seafood commodity and fishing country or flag state (referred to as countries). The SAU database builds from FAO global catch data using a bottom up, country and sector-specific approach that draws on grey literature and other sources to reconstruct catch patterns in each country. We limit our analysis to reported catch from industrial sectors, which are major suppliers of internationally traded seafood and tend to have more taxonomically detailed catch documentation. We repeated the analysis using a second global catch database also built from FAO catch data[60] (Supplementary Table 7 and Supplementary Fig. 1). We excluded one species, *Coregonus lavaretus*, because it exclusively inhabits freshwater ecosystems. There were more species-level catch records in the SAU database, but overall the patterns of threatened species catch and fishing countries were similar, with the exception of China. China's total reported catch in the SAU database is more than double any other fishing country, but the 2014 volume is likely an overestimate because it is derived from reconstructed catch estimates during a period of enormous expansion enabled by massive subsidies[1,61].

We then used a global seafood trade database to estimate the volume of international imports of each seafood commodity across importing countries, our best estimate of where the species is consumed[30]. The seafood trade database builds a virtual marketplace that links FAO FishStat Exports data to the fisheries catch. Country catches are matched to FAO FishStat exports records using the best approximations of taxa to commodity descriptions and data on bilateral trade partners from the United Nation's International Trade Statistics Database (UN ComTrade)[30]. The virtual marketplace identifies the source of the export (domestic catch, domestic aquaculture, foreign fishing, or re-exported product), and categorizes all non-matching exports or problematic import records as a re-export. Internationally traded seafood is difficult to trace through complex loops of importation, processing, and re-exportation as a different product, especially by major processors such as China[30]. We considered each country's catch and imports, excluding re-exported trade and aquaculture records.

Species biomass and fishing effort fluctuate considerably across years, so we selected the most recent decade in the databases (2006–2014 for catch and 2006–2015 for imports) to understand broad trends in fishing and seafood trade. To compare trends across threatened species and fishing or importing countries, we calculated weighted moving averages (WMAs) with 8- and 9-year windows for the most recent year (2014 and 2015, respectively). The WMA gives greater weight to more recent years by multiplying each value by a weighting factor. It is a common metric for forecasting data because it better represents trends compared to a simple average or total values.

Catch and imports are recorded as tonnes, underrepresenting the importance of small-bodied or rare species. We used ex-vessel price data from SAU to compare the economic value of threatened fish and invertebrates to industrial fisheries and to better represent low-volume but higher value species. The SAU database uses available price records to derive average ex-vessel prices (the price the fishers receive when they sell their landed catch), adjusted to USD, for all species-specific and non-species-specific commodities in the global catch database for each fishing country and year from 1950 to 2010[62].

Catch value is the product of volume and ex-vessel price for each commodity, country, and year. The price paid at the dock is often far less than the price of a highly processed commodity (e.g. breaded fillets) at its final import destination, but we use ex-vessel price to compare import value as well as catch value because it provides a data-driven metric of relative value for each species and commodity at a global scale.

**Statistical tests**. We posed two hypotheses about the key countries driving catch and trade of threatened species in industrial fisheries: (1) the world's major fishers and importers of all seafood commodities are the same countries that catch and import the largest volumes of threatened species, and (2) countries with better taxonomic resolution in their catch and import records will have larger volumes of threatened species recorded. To explore these questions, we used multiple linear regression models of threatened species catch and import volumes compared to the total volumes, and to the volumes of other record types (e.g. aggregated records). We tested per capita GDP and composite governance score as predictor variables using World Bank data accessed via the WDI and wbstats packages in R.

**Reporting summary**. Further information on research design is available in the Nature Research Reporting Summary linked to this article.

## Data availability

Two publicly publicly available databases were used in this study: (1) IUCN Red List of Threatened Species (https://www.iucnredlist.org) and (2) the RAM stock legacy database (https://www.ramlegacy.org/database/). We also used three private databases that have been published previously, and the full databases are available upon request. We provide the subsets of the private fisheries catch and trade data that needed to reproduce the results and figures as csv files in the public GitHub repository (https://github.com/lroberson/thr_seafood_pub). In addition, we requested the associated threat codes information for the selected threatened species from the IUCN Red List, and provide these data in the GitHub repository.

## Code availability

Analyses were conducted in R and the code used to produce the figures and tables is provided in R Markdown files in a public GitHub repository (https://github.com/lroberson/thr_seafood_pub).

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

## Acknowledgements

L.A.R. and C.J.K. are supported by the Centre for Biodiversity and Conservation Science at the University of Queensland. C.J.K. is supported by a University of Queensland Postdoctoral Fellowship. We thank James E. Watson for comments that improved this work, Maria Deng Palomares and Beth Polidoro for assisting with data access during a chaotic time, and Stephane Guillou, Awais Hameed Khan, Jarren Simmons, and Dan Vallentyne for assistance conceptualising and designing the figures for this manuscript.

## Author contributions

L.A.R. and C.J.K. conceptualized the project. R.A.W. created the global catch and trade databases and contributed to the analysis and interpretation of the data. L.A.R. analysed the data and prepared the figures. L.A.R. wrote the text, with substantial editorial input from C.J.K. All authors made editorial contributions to the final text.

## Competing interests

The authors declare no competing interests.
