## [Peer Review File · Nature Communications]

Reviewers' Comments:

Reviewer #1:

Remarks to the Author:

The major claim of the paper is that somewhere around 5% of reported catches from FAO data are Red Listed as a species of conservation concern (Vulnerable, Endangered, Critically Endangered). I appreciate the deep dive into somewhat messy data sources to provide insight into the impact of fisheries on fish and invert species of conservation concern. As an active participant in the IUCN SSC, I also appreciated the integration of Red List classifications.

Here are my primary concerns and some of these concerns may prove problematic to the central thesis of the manuscript:

- a) The language and organization/flow need modification as it is hard to follow. The language is unclear as to which of the fish/invert species that you are focusing on (in the 60) are targeted or commercially important species. There are a lot of percentages presented but it's not clear what these tell us or how important they are. Language/wording needs to be tightened, and more concise. e.g., very obvious gap there is no verb on the #2 listed in the final sentence of the intro.
- b) I found the thesis somewhat simplistic. Species with one of the 3 conservation classifications make up at most %5 of catch. With the exception of bluefin species, these species are not being targeted, but are reported in catch statistics. I also found the use of global vs. regional assessments to be problematic. I appreciate the challenges of deciding which spatial scale is "correct" but feel that this decision is to force convergence when there might not be. If this classification decision were reversed, it seems unlikely the following conclusion would stand "Four wide-ranging bony fish (haddock, bigeye tuna, Atlantic horse mackerel, and Madeiran sardinella) account for over 90% of threatened species catch" as only bigeye tuna would likely remain as threatened species.
- c) The assertion that harvest of species of conservation concern is not tolerated outside of fisheries is not a strong argument. There have been clear calls and efforts to "harvest" elephants (allow trophy hunting) and similar ongoing activities to harvest whales (see IWC) with the justification being that in some locations those populations are increasing or stable and the funds generated from the harvests can support management and conservation.

Reviewer #2:

Remarks to the Author:

General:

Overall, I consider the questions being asked in this study to be of considerable interest and relevance to the general community and wide readership of Nature Communications, and it should be of extreme relevance to the fisheries science community, whose perspective and outlook on fisheries needs to change. This manuscript meets these requirements, and is thus suitable for this journal.

However, I have one major concern and a few minor concerns that would need to be addressed before I would suggest acceptance by Nature Communications.

Main concern:

My main concern relates to some of the data sources being used here. I have no concerns with the IUCN data or the trade data and how they were filtered for use, this seems reasonable and logical, despite the extreme uncertainty and non-transparency (poor resolution) in the trade data, which is well articulated and acknowledged in the manuscript. My concern relates to the catch and ex-vessel price data used.

Lines 374-383: The catch data used here combine reported landed catches from official sources with so-called IUU data. However, the method used to estimate and add IUU catches is outdated and antiquated, and does not represent the newest stand of knowledge and data globally available. Overall, the data sources and methods used to assemble the data set used here for catches (Watson and Tidd 2018, *Marine Policy* 93: 171-177), while once representing an excellent first-order effort to improve on the minimalist FAO data, is now antiquated, non-comprehensive and non-transparent in light of other data sets now globally available. Since 2016, a far more comprehensive and detailed global data set in space and time is available that provides far more detailed and extensively documented and standardized data on global catches that carefully combine official reported records with best estimates of unreported (IUU) catches for every country in the world. This database, published by the Sea Around Us (www.searoundus.org) and documented in Pauly and Zeller (2016a, *Nature Communications* 7: 10244) and Pauly and Zeller (2016b, *Global Atlas of Marine Fisheries: A critical appraisal of catches and ecosystem impacts*. Island Press, 486 p.) should be used for this study.

Thus, the currently used data and data-methods are conceptually and content-wise outdated. The analysis needs to be repeated with the global catch data of the Sea Around Us.

Besides the above identified problematic IUU (unreported) catch estimates used here, equally important in this context is the manner in which the currently used catch data tries to differentiate and sub select catch data in terms of fisheries sectors, i.e. "industrial" versus "non-industrial". The differentiation used in Watson and Tidd (2018) and thus in the present study is outdated, inaccurate and problematic because it uses global proxies rather than local, country specific sector differentiation. Again, the data of the Sea Around Us has bottom-up country by country reconstructed data that are sector-by-country specific, making for the only comprehensive data set in the world that can differentiate between industrial and non-industrial (see also Pauly and Charles 2015, *Science* 347: 242-243). This is a second major reason why the present study and analysis needs to be repeated using Sea Around Us catch data instead of the currently used, antiquated data.

Lines 397-417: Similarly, the ex-vessel price data used here (apparently the 2007 version) is unfortunately also outdated, and several updates exist. One such update is even referenced in the manuscript (Swartz et al. 2015) but surprisingly is not used as data? Instead, a proxy inflation-rate adjustment to the older, outdated data set of 2007 is used. This is a flagrant and knowing misuse of outdated data that cannot stand for a journal of the caliber of *Nature Communications*. Furthermore, even a superficial web search reveals that the ex-vessel price data seem to have been updated again since 2015, see Tai et al. (2017, *Frontiers in Marine Science* 4: 363). Thus, in order to be relevant and appropriate for this journal, the analysis has to be repeated with the latest 2017 ex-vessel price data as documented in Tai et al. (2017).

Minor comments:

Throughout the manuscript: Regular use of the term "legal" fishing is made in direct reference to the reported catch data being used (e.g. Lines 67, 113, 116, 183, 215, 383 etc.). This is misleading and inaccurate. Only because catch data are reported or not reported does not mean they are legal or illegal, as they may be illegally caught yet end up in reported records. Equally, unreported catches (i.e., part of IUU) may be perfectly legally caught, just not reported. All it means is that these catches are reported... or unreported. Much illegal, as well as legal catch is known to be transhipped, yet still enters either the legal or illegal landings and/or trade systems. Thus, remove all reference to "legal" and "illegal" from the manuscript, this paper is not about legal versus illegal, only about reported/unreported catches and traded catches, with no reference to legally.

Line 137 and 139: The significance of Morocco needs to be better highlighted and explained. It is unlikely that Morocco has such a large truly national industrial fleet to result in global significance in IUCN listed catches. Could this not relate more to the use of its flag as flag of convenience, and thus

does not truly represent Morocco? This needs to be far more deeply investigated, highlighted and discussed. Similar concerns relate to Northern Mariana. This also needs far more detailed examinations and explanation.

Line 376: Why is "spatially-explicit" relevant? The analysis conducted here seems to be global by fishing country (see Supplementary Table S2), so space or location of catches is irrelevant here.

Line 378: In the same manner as space is irrelevant in the present analysis, the GFW and AIS data is irrelevant here and need no mentioning. Given that the analysis needs to be repeated with the comprehensive Sea Around Us data, these items become unnecessary anyway, as all this methods text needs removing during methods rewrite.

Lines 414-417: The authors should seriously consider making use of the economic multipliers described in Dyck and Sumaila (2010, *Journal of Bioeconomics* 12: 227-243), as these may be of utility in terms of value added evaluation.

Point-by-point response to reviewer comments (reviewer comments are in italics).

Reviewer #1 (Remarks to the Author):

The major claim of the paper is that somewhere around 5% of reported catches from FAO data are Red Listed as a species of conservation concern (Vulnerable, Endangered, Critically Endangered). I appreciate the deep dive into somewhat messy data sources to provide insight into the impact of fisheries on fish and invert species of conservation concern. As an active participant in the IUCN SSC, I also appreciated the integration of Red List classifications.

Here are my primary concerns and some of these concerns may prove problematic to the central thesis of the manuscript:

a) The language and organization/flow need modification as it is hard to follow. The language is unclear as to which of the fish/invert species that you are focusing on (in the 60) are targeted or commercially important species. There are a lot of percentages presented but it's not clear what these tell us or how important they are. Language/wording needs to be tightened, and more concise. e.g., very obvious gap there is no verb on the #2 listed in the final sentence of the intro.

1. To increase clarity around percentages presented, we removed many of the unnecessary statistics (especially in the "Country level patterns in catch and imports" paragraph, **line 142**). We also removed several statistics from the paragraphs about catch (**line 82**) and imports (**line 124**) results.

2. We have made substantial edits focusing on flow and organization throughout the manuscript. For example, we moved the paragraph, "While we have yet to fish a widely abundant marine fish or invertebrate species to extinction..." to the introduction (**line 46**), to improve the flow from the discussion of targeting in the previous paragraph. We also made substantial edits to the paragraph about threats data (**line 97**), in line with response 1 regarding statistics and organization. Many other changes associated with flow and organization are related to other comments and are discussed in more detail below. Overall, we believe the revised manuscript is easier to follow. We cut more than 500 words from the text (including the Methods section). We have also fixed the final sentence in the introduction, as suggested (**line 76**).

Thank you for the suggestion regarding targeting in commercial sectors, we address this point in the following comment.

b) I found the thesis somewhat simplistic. Species with one of the 3 conservation classifications make up at most %5 of catch. With the exception of bluefin species, these species are not being targeted, but are reported in catch statistics. I also found the use of global vs. regional assessments to be problematic. I appreciate the challenges of deciding which spatial scale is "correct" but feel that this decision is to force convergence when there might not be. If this classification decision were reversed, it seems unlikely the following conclusion would stand "Four wide-ranging bony fish (haddock, bigeye tuna, Atlantic horse mackerel, and Madeiran sardinella) account for over 90% of threatened species catch" as only bigeye tuna would likely remain as threatened species.

3. This is a very useful suggestion and an important topic. The FAO does not categorize species as target versus incidental in the global catch data, and there is no global list of commercially targeted species, so an in-depth analysis of what is targeted and what is incidental is difficult. However, we were able to address this by using the IUCN threats data in combination with the RAM Stock Legacy database, which compiles stock assessment results

for commercially exploited marine fish and invertebrates around the world (<https://www.ramlegacy.org/>). Although the curators acknowledge that many assessed stocks (or entire countries with assessed stocks) are missing from the database, to our knowledge, it is the best available global indicator of whether a species is commercially targeted (or at least one stock of that species). We found that 27 of the 61 threatened species from the original catch data are commercially targeted, and 34 of the 92 species from the Sea Around Us data (data used for a new analysis, discussed in later comments) are listed as commercial targets in the RAM database. The IUCN threats data lists an additional 39 species as threatened with large-scale, intentional fishing. We redesigned Figure 1, which now shows the RAM-listed commercially targeted species in bold, and discussed this addition in line 106. We have added two sentences about non-target catch documentation in the discussion (**lines 274-277**).

4. Regarding the regional assessments, we agree that it is confusing to discuss both spatial scales. However, it is important to acknowledge that regional stocks can have a different threat status than indicated in global assessments of the sample species. We removed the regional assessment information in Supplementary Table 1 for all species. However, given the ongoing debate about the Atlantic cod's particularly controversial global IUCN status, we have decided to use its regional assessment (Least Concern), as we explain in more detail in lines (**lines 327-332**). In line with your comment, we rewrote this paragraph in the Methods (**line 321**).

c) The assertion that harvest of species of conservation concern is not tolerated outside of fisheries is not a strong argument. There have been clear calls and efforts to "harvest" elephants (allow trophy hunting) and similar ongoing activities to harvest whales (see IWC) with the justification being that in some locations those populations are increasing or stable and the funds generated from the harvests can support management and conservation.

5. This is an interesting point indeed and an important topic to discuss. In response to this comment, we have expanded our discussion to better capture the issues raised regarding the comparison between fishing and hunting or harvest of charismatic species. Prior to this, we only briefly mentioned it, which did not do it justice. Our point is that industrial fishing (for food) is unique because it is "the world's last major hunting and gathering industry" (Asche, F., et al. 2018, PNAS). As a result, we used the words, "harvest," "fish," and "hunt" somewhat interchangeably throughout the manuscript, which could generate some confusion about the manuscript's overarching aim. Our aim is to explore reported, sanctioned, industrial-scale harvest of species listed as threatened with extinction. To be more consistent with this aim and address this comment, we changed the wording at various points in the manuscript (e.g. line 13 of the abstract, "industrial-scale harvesting of species at risk of extinction..." instead of, "hunting species at risk..."). We also changed **lines 39-41** in the introduction and added **lines 41-44** to specifically address the Reviewer's comments about trophy hunting and whaling.

Reviewer #2 (Remarks to the Author):

General:

Overall, I consider the questions being asked in this study to be of considerable interest and relevance to the general community and wide readership of Nature Communications, and it should be of extreme relevance to the fisheries science community, whose perspective and outlook on fisheries needs to change. This manuscript meets these requirements, and is thus suitable for this journal. However, I have one major concern and a few minor concerns that would need to be addressed before I would suggest acceptance by Nature Communications.

Main concern:

My main concern relates to some of the data sources being used here. I have no concerns with the IUCN data or the trade data and how they were filtered for use, this seems reasonable and logical, despite the extreme uncertainty and non-transparency (poor resolution) in the trade data, which is well articulated and acknowledged in the manuscript. My concern relates to the catch and ex-vessel price data used.

Lines 374-383: The catch data used here combine reported landed catches from official sources with so-called IUU data. However, the method used to estimate and add IUU catches is outdated and antiquated, and does not represent the newest stand of knowledge and data globally available. Overall, the data sources and methods used to assemble the data set used here for catches (Watson and Tidd 2018, Marine Policy 93: 171-177), while once representing an excellent first-order effort to improve on the minimalist FAO data, is now antiquated, non-comprehensive and non-transparent in light of other data sets now globally available. Since 2016, a far more comprehensive and detailed global data set in space and time is available that provides far more detailed and extensively documented and standardized data on global catches that carefully combine official reported records with best estimates of unreported (IUU) catches for every country in the world. This database, published by the Sea Around Us (www.searoundus.org) and documented in Pauly and Zeller (2016a, Nature Communications 7: 10244) and Pauly and Zeller (2016b, Global Atlas of Marine Fisheries: A critical appraisal of catches and ecosystem impacts. Island Press, 486 p.) should be used for this study. Thus, the currently used data and data-methods are conceptually and content-wise outdated. The analysis needs to be repeated with the global catch data of the Sea Around Us.

6. We have addressed this comment by repeating the analysis with the Sea Around Us catch database. The Sea Around Us database has greatly advanced our understanding of global fishing patterns, and we believe incorporating these data strengthens the manuscript. We have included the results of this analysis in the main text. Analysis using the Watson & Tidd data were included in the supplementary materials. We felt that including them both made the paper more robust as there were several similarities in the results. Further, each database has different limitations; they are limited by data that is missing, misreported, or low-resolution, and both sources acknowledge these limitations and attempt to fill gaps in different ways.

In the revised manuscript, we repeat the analysis with the SAU catch database and with the newer ex-vessel prices derived from this data, as explained in a subsequent comment. All 61 threatened species reported in the Watson & Tidd database also appeared in the SAU catch data, along with 31 additional threatened species. Importantly, the higher taxonomic resolution can be attributed to SAU's incorporation of country-specific reports into their catch estimates, which you highlight. The fishing country results are similar, although SAU estimates for China's total catch are much higher than those in Watson & Tidd. We discuss this in **lines 356-358**. The SAU data corroborates our previous findings that European countries and a handful of Asian and West African nations have the highest volumes and value of threatened species catch, although the order of rankings changed slightly.

7. Indeed, you make an important point that the Watson & Tidd data does not focus on documenting IUU-associated catch, whereas the Sea Around Us data includes more detailed methods for characterizing this component of fisheries catches. As our study focuses only on reported catch in larger-scale fisheries, we excluded IUU-associated catch from the analysis (**line 351**). However, in the original analysis we did include it as an explanatory variable in the GLMs. To address this important comment, we removed this from the models and from the analysis entirely, as it could be confusing and it is superfluous to our analysis (**see revised Methods, lines 393-400**).

Besides the above identified problematic IUU (unreported) catch estimates used here, equally important in this context is the manner in which the currently used catch data tries to differentiate and sub select catch data in terms of fisheries sectors, i.e. "industrial" versus "non-industrial". The differentiation used in Watson and Tidd

(2018) and thus in the present study is outdated, inaccurate and problematic because it uses global proxies rather than local, country specific sector differentiation. Again, the data of the Sea Around Us has bottom-up country by country reconstructed data that are sector-by-country specific, making for the only comprehensive data set in the world that can differentiate between industrial and non-industrial (see also Pauly and Charles 2015, *Science* 347: 242-243). This is a second major reason why the present study and analysis needs to be repeated using Sea Around Us catch data instead of the currently used, antiquated data.

8. We agree that differentiating between industrial and non-industrial sectors is difficult and problematic across all catch databases. The SAU database places greater emphasis on defining the fishing sector (e.g. they include artisanal, industrial, recreational, and subsistence categories). In contrast, the database described in Watson & Tidd (2018) includes only two categories, industrial and non-industrial, but the non-industrial records are negligible (only 0.1% percent of global reported landings volume during our time frame). To address this comment, we repeated the analysis with the SAU catch data. We exclude recreational, artisanal, and subsistence sectors for which there is less oversight and more decentralized management compared to larger sectors, and importantly, where catch data is often lacking and has been reconstructed (Chuenpagdee et al. 2006, *Fisheries Centre Research Reports* 14; Ruttan et al. 2000, *Sea Around Us Project Methodology Review*).

The aim of our manuscript is to make the most conservative estimate of threatened species catch in large-scale, industrial fisheries (e.g. by selecting only species-level records), thus, we limit the analysis to "industrial" catch to focus on fisheries with higher taxonomic resolution in the raw catch data. We compared reported "industrial" catch from both databases, as described in the previous comment, and find similar results across fishing countries (**Supplementary Table 7 and Supplementary Figure 1**).

*Lines 397-417: Similarly, the ex-vessel price data used here (apparently the 2007 version) is unfortunately also outdated, and several updates exist. One such update is even referenced in the manuscript (Swartz et al. 2013) but surprisingly is not used as data? Instead, a proxy inflation-rate adjustment to the older, outdated data set of 2007 is used. This is a flagrant and knowing misuse of outdated data that cannot stand for a journal of the caliber of Nature Communications. Furthermore, even a superficial web search reveals that the ex-vessel price data seem to have been updated again since 2015, see Tai et al. (2017, *Frontiers in Marine Science* 4: 363). Thus, in order to be relevant and appropriate for this journal, the analysis has to be repeated with the latest 2017 ex-vessel price data as documented in Tai et al. (2017).*

9. We agree that it is important to use the best available price data. To address this, we redid our analysis using the suggested data, described in Tai et al. 2017. The authors of this data advised us that the best way to obtain the information is to calculate it from the Sea Around Us data (dividing value by landings for each commodity), which we have done in the revised manuscript. The lead author also informed us that the price data only extends to 2010, and the catch values for the years 2011-2014 in the Sea Around Us data are calculated using the 2010 prices. As 2010 is in the middle of our time period (2006 - 2014/5), we believe it is an appropriate value to use and agree that it is better than attempting to extrapolate prices beyond the dataset.

We did conduct a second search for more recent ex-vessel price data, but the data described in Tai et al. 2017 is the only dataset with global coverage and resolution at the level of individual commodities/species. For example, there are more recent prices given in Melnychuk et al. 2017, but they are in aggregated groups (e.g. "tunas") and thus would not match our data (Melnychuk, M. C., Clavelle, T., Owashi, B. & Strauss, K. Reconstruction of global ex-vessel prices of fished species. *ICES J. Mar. Sci.* 74, 121–133 (2017)). Swartz et al. 2013 updates the methodology for filling missing prices in the 2007 data, but unfortunately does not update the prices themselves.

Minor comments:

Throughout the manuscript: Regular use of the term "legal" fishing is made in direct reference to the reported catch data being used (e.g. Lines 67, 113, 116, 183, 215, 383 etc.). This is misleading and inaccurate. Only because catch data are reported or not reported does not mean they are legal or illegal, as they may be illegally caught yet end up in reported records. Equally, unreported catches (i.e., part of IUU) may be perfectly legally caught, just not reported. All it means is that these catches are reported... or unreported. Much illegal, as well as legal catch is known to be transshipped, yet still enters either the legal or illegal landings and/or trade systems. Thus, remove all reference to "legal" and "illegal" from the manuscript, this paper is not about legal versus illegal, only about reported/unreported catches and traded catches, with no reference to legally.

10. Thank you for pointing this out, you are correct and it is important not to confuse these two terms. We have changed all instances of "legal" to "reported," including in the title of the manuscript.

Line 137 and 139: The significance of Morocco needs to be better highlighted and explained. It is unlikely that Morocco has such a large truly national industrial fleet to result in global significance in IUCN listed catches. Could this not relate more to the use of its flag as flag of convenience, and thus does not truly represent Morocco? This needs to be far more deeply investigated, highlighted and discussed. Similar concerns relate to Northern Mariana. This also needs far more detailed examinations and explanation.

11. This is a good point. The Reviewer is correct that flags of convenience are a key factor driving the large reported catches from Morocco and Northern Marianas, as well as licensing of distant water fleets (Belhabib et al. 2016, Marine Policy; Zeller et al. 2015, Coral Reefs; Miller et al. 2014, Marine Policy). To address this, we removed that sentence from the results as it is somewhat superfluous and merits an in-depth discussion that is beyond the scope of this paper. We do discuss flags of convenience in the discussion (**lines 279-283**), as it is a critical factor affecting country-level catch data. Our objective is not to point fingers at specific countries for catching more threatened species, as there are many context-specific dynamics that we do not discuss in this global analysis. Rather, we aim to broadly highlight patterns in reported catch and trade of threatened species across countries, and, importantly, the large variations in documentation and record quality.

Line 376: Why is "spatially-explicit" relevant? The analysis conducted here seems to be global by fishing country (see Supplementary Table S2), so space or location of catches is irrelevant here.

12. We appreciate this suggestion as it should help clarify our methods. In the original manuscript we provided detailed information about the global catch database, but much of this information is extraneous to our analysis (e.g. the spatial information about how the catch was derived). We have removed this text and believe it makes the methods easier to follow, especially since these methods are all provided in Watson & Tidd 2018. The same applies to the categorization of industrial versus nonindustrial sectors, and IUU fishing, which we discussed in **responses 7 and 8**.

Line 378: In the same manner as space is irrelevant in the present analysis, the GFW and AIS data is irrelevant here and need no mentioning. Given that the analysis needs to be repeated with the comprehensive Sea Around Us data, these items become unnecessary anyway, as all this methods text needs removing during methods rewrite.

13. We have removed the information about the derivation of the catch data from the Methods to make this section more concise, as suggested in the previous comment.

Lines 414-417: The authors should seriously consider making use of the economic multipliers described in Dyck

and Sumaila (2010, Journal of Bioeconomics 12: 227-243), as these may be of utility in terms of value added evaluation.

14. Thank you for bringing this paper to our attention. It would be an interesting extension of our study to explore the economic importance of threatened fisheries species to the fishing and importing countries. For example, the economic multipliers described in Dyck and Sumaila (2010) would be useful for an analysis of catch volume and value in the context of each country's fishing industry, and compared to their economy more generally. We thought about how we might incorporate these elements into our manuscript, but decided it might distract from the main objectives of our analysis. The objective of including the price data was primarily to provide another perspective on catch patterns across species, in addition to volume. For example, it highlights comparisons between species with very high value but low catch volume (e.g. lobster) to a high volume but lower value species (e.g. haddock).

Reviewers' Comments:

Reviewer #1:

Remarks to the Author:

The authors thoroughly addressed reviewer comments and I am satisfied with the changes

Reviewer #2:

Remarks to the Author:

Re-analysing the paper with Sea Around Us data and incorporating the other reviewer comments has substantially improved the manuscript. It is more internally comprehensive and consistent, and is now based on better data. Note that the PDF file for reviewers contained two sets of the MS, one after the other. I assume this was an error and that both versions are identical. I only reviewed the first iteration of the MS.

I would like to add two points here that relate to comments made by the other reviewer with regards to 'target' versus 'incidental' catch of species. While indeed in many developed countries there exist so-called 'target' fisheries that aim to mainly catch a certain species (e.g. North Sea herring fisheries), or set of species, this is less so the case in most places in the world, i.e. outside of the developed world. Thus, 'global' fisheries are far less 'targeting' than one may think, and instead catch whatever they can catch, and then retain whatever they can sell. Thus, I am less concerned about the issues around 'target' versus 'incidental' as my reviewer colleague. Furthermore, there are now fisheries, such as the bottom fish fisheries off the west coast of Canada in which catches and fisher behaviour are increasingly driven by avoidance behaviour of fishers, to avoid catching species that have either very low quotas or are not permitted to be caught. Thus, fisheries maximize avoidance of species in such situations rather than maximizing any 'target' quotas, as otherwise their fisheries can close down without any 'target' species quotas being filled. I think this is the right thinking, and far more fisheries around the world should think as such and implement such rules for endangered or threatened species.

Minor comments:

Line 21: 'tons' is incorrect here, as all data are in 'tonnes', or 'metric tons' for US-only readers. 'Tonnes' is the international scientific standard and should be used throughout. Later in the MS the authors correctly indicate that all data are in 'tonnes'. Change here to 'tonnes'.

Supplementary tables SI Table 1, SI table 3 and SI table 7: Species names need to be italicized.

REVIEWERS' COMMENTS:

Reviewer #1 (Remarks to the Author):

The authors thoroughly addressed reviewer comments and I am satisfied with the changes

Reviewer #2 (Remarks to the Author):

Re-analysing the paper with Sea Around Us data and incorporating the other reviewer comments has substantially improved the manuscript. It is more internally comprehensive and consistent, and is now based on better data. Note that the PDF file for reviewers contained two sets of the MS, one after the other. I assume this was an error and that both versions are identical. I only reviewed the first iteration of the MS.

I would like to add two points here that relate to comments made by the other reviewer with regards to 'target' versus 'incidental' catch of species. While indeed in many developed countries there exist so-called 'target' fisheries that aim to mainly catch a certain species (e.g. North Sea herring fisheries), or set of species, this is less so the case in most places in the world, i.e. outside of the developed world. Thus, 'global' fisheries are far less 'targeting' than one may think, and instead catch whatever they can catch, and then retain whatever they can sell. Thus, I am less concerned about the issues around 'target' versus 'incidental' as my reviewer colleague. Furthermore, there are now fisheries, such as the bottom fish fisheries off the west coast of Canada in which catches and fisher behaviour are increasingly driven by avoidance behaviour of fishers, to avoid catching species that have either very low quotas or are not permitted to be caught. Thus, fisheries maximize avoidance of species in such situations rather than maximizing any 'target' quotas, as otherwise their fisheries can close down without any 'target' species quotas being filled. I think this is the right thinking, and far more fisheries around the world should think as such and implement such rules for endangered or threatened species.

Thank you for the insightful comments about targeting in industrial fisheries and the eloquent summary of this problem. We agree that more fisheries should shift their thinking towards encouraging avoidance behaviors of endangered species.

Minor comments:

Line 21: 'tons' is incorrect here, as all data are in 'tonnes', or 'metric tons' for US-only readers. 'Tonnes' is the international scientific standard and should be used throughout. Later in the MS the authors correctly indicate that all data are in 'tonnes'. Change here to 'tonnes'.

Thank you for catching this error, we have corrected it.

Supplementary tables SI Table 1, SI table 3 and SI table 7: Species names need to be italicized.

Corrected.